

# Global inverse modeling of CH₄ sources and sinks: An overview of methods

Sander Houweling[1,2], Peter Bergamaschi[3], Frederic Chevallier[4], Martin Heimann[5], Thomas Kaminski[6], Maarten Krol[1,2,7], Anna M. Michalak[8], and Prabir Patra[9]

[1]SRON Netherlands Institute for Space Research, Utrecht, The Netherlands.
[2]Institute for Marine and Atmospheric Research (IMAU), Utrecht University, Utrecht, The Netherlands.
[3]European Commission Joint Research Centre, Institute for Environment and Sustainability, Ispra (Va), Italy.
[4]Le Laboratoire des Sciences du Climat et l'Environnement (LSCE), Gif-Sur-Yvette, France.
[5]Max-Planck-Institute for Biogeochemistry, Jena, Germany.
[6]The Inversion Lab, Hamburg, Germany.
[7]Department of Meteorology and Air Quality (MAQ), Wageningen University and Research Centre, Wageningen, The Netherlands.
[8]Department of Global Ecology, Carnegie Institution for Science, Stanford, U. S. A.
[9]Japanese Agency for Marine-Earth Science and Technology, Yokohama, Japan.

*Correspondence to:* Sander Houweling (s.houweling@sron.nl)

**Abstract.** The aim of this paper is to present an overview of inverse modeling methods that have been developed over the years for estimating the global sources and sinks of CH₄. It provides insight into how techniques and estimates have evolved over time, and what the remaining shortcomings are. As such, it serves a didactical purpose of introducing apprentices to the field, but it also takes stock of the developments so far and reflects on promising new directions. The main focus is on methodological

5    aspects that are particularly relevant for CH₄, such as its atmospheric oxidation, the use of methane isotopologues, and specific challenges in atmospheric transport modeling of CH₄. The use of satellite retrievals receives special attention, as it is an active field of methodological development, with special requirements on the sampling of the model and the treatment of data uncertainty. Regional scale flux estimation and attribution is still a grand challenge, which calls for new methods capable of combining information from multiple data streams of different measured parameters. A process model representation of

10    sources and sinks in atmospheric transport inversion schemes allows the integrated use of such data. These new developments are needed not only to improve our understanding of the main processes driving the observed global trend, but also to support international efforts to reduce greenhouse gas emissions.

## 1 Introduction

Thanks to the efforts of surface monitoring networks, the global trends of long-lived greenhouse gases over the past decades

15    are known to high accuracy (Dlugokencky et al., 2009; Prinn et al., 2000; Simpson et al., 2012). However, deciphering the causes of observed growth rate variations remains a challenge, and is an active field of scientific research and development. The large variations in the methane growth rate that have been observed in the past years are a particularly good example. A wide variety of possible scenarios have been discussed in the recent literature, but only limited consensus has been reached



so far (Rigby et al., 2008; Bousquet et al., 2011; Monteil et al., 2011; Kai et al., 2011; Aydin et al., 2011; Bergamaschi et al., 2013; Houweling et al., 2014; Turner et al., 2016; Schaefer et al., 2016; Patra et al., 2016; Franco et al., 2016).

The reason for which the origin of these growth rate variations is difficult to identify was already discussed extensively in the late 1980s to early 1990s, when the first inverse modeling techniques were developed for inferring greenhouse gas sources and sinks from atmospheric measurements (Newsam and Enting, 1988; Enting and Newsam, 1990). The inverse problem was qualified as "ill posed", because of the wide range of surface flux configurations that could explain the measurements about equally well. Such problems require regularization using *a priori* assumptions on the surface fluxes, needed to fill in critical flux information that the measurement networks are unable to provide.

Since then several approaches have been investigated to strengthen the constraints brought in by the measurements, for example, by increasing the amount of data using regional tall tower networks (Bergamaschi et al., 2015; Miller et al., 2013) and satellites (Meirink et al., 2008a; Monteil et al., 2013; Cressot et al., 2014; Wecht et al., 2014) or by using different types of measurements, including methane isotopologues (Mikaloff Fletcher et al., 2004a; Pison et al., 2009; Neef et al., 2010). To accommodate new kinds of data in the inversion framework posed new methodological challenges, not only the computational challenge of solving an inverse problem of significantly increased size, but also the treatment of new measurements with poorly quantified error statistics (Houweling et al., 2014). Then, with improved measurement capabilities increasing the flux resolving power of the inversions, transport model uncertainties were recognized to play an increasingly important role (Patra et al., 2011; Kirschke et al., 2013).

Despite methodological limitations, the inverse modeling approach allowed us to derive important constraints on the global sources and sinks of $CH_4$. Examples are the dominant role of the tropical and temperate northern latitudes as drivers of the observed methane increase since 2007 (Bousquet et al., 2011). These constraints exist despite the limited availability of surface measurements in the Tropics. The extension of global inversions with satellite retrievals from SCIAMACHY and GOSAT confirmed and even reinforced the importance of tropical fluxes (Frankenberg et al., 2005; Beck et al., 2012; Wilson et al., 2016). Initially, using SCIAMACHY, it took a correction to account for an overestimated role of the Tropics due to spectroscopic errors affecting the $XCH_4$ retrieval (Frankenberg et al., 2008). For the Boreal and Arctic latitudes, inversions confirm the sensitivity of methane fluxes to climatic variability, however, as yet without significant trends in response to global warming (Bergamaschi et al., 2013; Bruhwiler et al., 2014). Regarding the atmospheric sink strength, inversions have put bounds on the plausible range of OH inter-annual variability, although it remains difficult to quantify surface sources and atmospheric sinks independently of each other using the available measurements (Rigby et al., 2008).

The purpose of this paper is to review methods in global inverse modeling of $CH_4$, and directions in which the field is developing. The discussion is limited mostly to global and contemporary methane, although the range of applications has expanded over the years, covering scales ranging from paleoclimate studies (Fischer et al., 2008) to the estimation of single point sources (Kort et al., 2014). Inverse modeling of $CH_4$ has taken advantage of methodological advances gained in the application of inverse modeling to $CO_2$, except for some aspects that are specific to $CH_4$, such as its limited atmospheric lifetime, which will receive special attention.





The next section starts with an overview of how $CH_4$ inversions evolved over the years. The treatment of atmospheric sinks is discussed separately in section 3. Sections 4 – 6 zoom in on the use of isotopic measurements, satellites, and the role of chemistry transport models. Finally, new developments and directions are discussed in section 7.

## 2   The evolution of methods and estimates

The first inverse modeling analyses of global $CH_4$ made use of concepts and techniques that were developed earlier for studying $CO_2$, as published for example by Enting (1985) and Enting and Mansbridge (1989). The first synthesis of global methane was performed by Fung et al. (1991), who assessed the contribution of various processes to the observed concentrations using a 3D atmospheric transport model. Sources were not yet optimized using an objective mathematical procedure. Instead, seven scenarios were presented that agreed with the available information on emissions and photochemical oxidation of methane as well as observed quantities, such as global mean $CH_4$, $^{13}CH_4$, and $^{14}CH_4$, the amplitudes of their seasonal cycles, and latitudinal gradients.

Brown (1993, 1995) was the first to apply a matrix inversion approach to the available background measurements to derive optimized monthly methane fluxes in 18 latitudinal bands. In Brown (1993) the number of unknowns was kept equal to the number of knowns in order to derive a unique solution. In the follow up study (Brown, 1995) this condition was relaxed, through the use of a truncated singular value decomposition approach. Both studies accounted for the atmospheric sink of methane by prescribing model calculated OH fields, which had been optimized to bring the global lifetime of methyl chloroform (MCF) in agreement with measurements (see for example Spivakovsky et al. (1990)).

Hein et al. (1997) followed the "synthesis inversion" concept of Enting (1993), which made use of a Bayesian formulation of the cost function penalizing deviations from a first guess (*a priori*) set of $CH_4$ fluxes. In this study, the state vector consisted of global and seasonal patterns of each source and sink process, as well as process-specific $\delta^{13}C$ isotopic fractionation factors. Houweling et al. (1999) relaxed the hard-constraint on global flux patterns by using the adjoint of the TM2 transport model, coded by Kaminski et al. (1996), to optimize the net $CH_4$ surface flux per month and at the resolution of the transport model. In addition, an attempt was made to take the spatial and temporal correlation of the uncertainty of the monthly fluxes between surrounding grid boxes into account. An iterative procedure was used to minimize the cost function, in order to account for the weak non-linearity introduced by optimizing the global OH sink (see section 3). In later studies flux regions have been defined in various ways, ranging between the global patterns of Hein et al. (1997) and the grid scale fluxes of Houweling et al. (1999), such as, for example, the use of 11 continental TransCom regions in Bousquet et al. (2006).

Up to this stage, inverse modeling studies had addressed multi-year mean sources and sinks, and their average seasonal variability. For example, the results of Hein et al. (1997) represented a quasi stationary state, reflecting the mean $CH_4$ increase during the analyzed time window of a few years, caused by the mean imbalance between the global sources and sinks during that period. Consistent with this approach, the atmospheric transport model recycled a single representative meteorological year. Important $CH_4$ growth rate fluctuations that were observed in the 1990s, such as in the years after the eruption of Mt. Pinatubo and during the strong 1997-1998 El-Niño, raised the interest in methods that could address inter-annual variability. To





do this, the use of actual meteorology in atmospheric transport modeling was recognized as being critical, since an important fraction of the observed inter-annual variability in $CH_4$ could be explained by variability in transport (Warwick et al., 2002).

The first so-called "time dependent" inversion of methane was published by Mikaloff Fletcher et al. (2004a), using the Kalman filter for the optimization of $CH_4$ fluxes (Bruhwiler et al., 2000). In this inversion, surface emissions were optimized

given a scenario for the sinks, i.e. without co-optimizing atmospheric sinks. This approach avoided spurious covariance between the inversion-optimized sources and sinks, resulting from the surface network providing insufficient information to constrain these terms independently. Later studies, such as Pison et al. (2009), introduced independent information about the sink, through the combined use of $CH_4$ and MCF measurements. Although this approach limits the trade-off between sources and sinks, some degree of influence remains depending on the weight of the $CH_4$ data relative to those of MCF. The weight of $CH_4$

data increased in particular with the use of satellite data. Several studies using SCIAMACHY satellite retrievals returned to the use of prescribed OH fields (Bergamaschi et al., 2007; Meirink et al., 2008b; Bergamaschi et al., 2009, 2013; Houweling et al., 2014).

The availability of satellite data, starting with the SCIAMACHY instrument onboard ENVISAT (Bovensmann et al., 1999), triggered new methodological developments to deal with the large amount of data becoming available, and to make efficient use

of the improved measurement coverage. Several groups adopted the 4D-VAR technique, developed by the weather prediction community, which makes use of the adjoint of the atmospheric transport model for efficient calculation of source receptor relationships and the cost function gradient (Rayner et al., 2016; Chevallier et al., 2005; Baker et al., 2006; Meirink et al., 2008b). With the increasing power of massively parallel super computers the ensemble Kalman filter (EnKF) gained popularity (Feng et al., 2009; Fraser et al., 2013; Peters et al., 2005; Bruhwiler et al., 2014).

To use satellite data, the inversions were extended with bias correction algorithms to account for systematic errors in the satellite retrievals. Various approaches were tested (see section 5) with spatiotemporally varying bias functions either optimized within the inversion or separately using measurements from the Total Column Carbon Observing Network (TCCON, Wunch et al. (2011a)). Because of known short-comings of atmospheric transport models, for example in simulating the stratosphere-troposphere exchange, inversion-optimized bias corrections were found to account in part for model deficiencies (Monteil et al.,

2013; Alexe et al., 2015; Locatelli et al., 2015). Compared with $CO_2$, stratosphere-troposphere exchange is relatively important for the column average mixing ratio of $CH_4$, because of the steeper vertical gradient of $CH_4$ in the stratosphere caused by its chemical transformation. The absence of $CH_4$ in the stratosphere matters, because concentration gradients provide the flux information that is used in inversions, and should therefore be represented well in models.

The proxy retrieval method, developed for the retrieval of $CH_4$ from SCIAMACHY (Frankenberg et al., 2005), has an

additional source of systematic error from the use of transport model output (Parker et al., 2015; Pandey et al., 2016). In this method, $XCH_4$ is derived from the satellite retrieved ratio of $XCH_4$ and $XCO_2$ to mitigate errors due to light scattering on cirrus and aerosol particles. To translate the retrieved ratios into $XCH_4$, model-derived estimates of $XCO_2$ are used. When proxy retrievals are used in inversions, inaccuracies in the modeled $XCO_2$ variations are projected on the $CH_4$ fluxes (Parker et al., 2015). To deal with this problem, dual $CO_2$ and $CH_4$ inversions have been developed, which directly assimilate satellite

retrieved ratios of $XCH_4$ and $XCO_2$ (Fraser et al., 2013; Pandey et al., 2015, 2016), together with surface measurements.



Figure 1 presents large-scale estimates from published global CH$_4$ inversion, and how they evolved over time. The global flux is the best constrained property, and shows reasonable consistency across the published studies. The range of estimates reflects mostly the improving capability to constrain the atmospheric oxidation of methane, which was still limited during the 1990s. Brown (1995) explicitly mentions that the difference with Brown (1993) is largely due to the choice of methane lifetime. Notice that the large error margin reported in Brown (1993) is consistent with this difference. Apart from this study, the inversion-derived estimates until 2006 cluster in two groups that differ by 80-100 TgCH$_4$/yr. The global flux estimates in more recent studies suggest that a consensus has been reached in favor of the lower cluster of estimates at 490–520 TgCH$_4$/yr for the 1990s. The increasing number of studies covering the period of renewed methane growth show an upward tendency consistent with the CH$_4$ increase. Note that these numbers are intended to include the soil sink, estimated to be in the range of 26–42 TgCH$_4$ (Kirschke et al., 2013), but it is not always clear if reported global emissions include or exclude this sink.

The contribution of the Northern Hemisphere to global emissions varies between 67 and 88% without a clear trend. The differences between the inversions may be explained largely by differences in the inter-hemispheric exchange rate of the transport models that are used (Patra et al., 2011). In Houweling et al. (1999) the exchange of the TM2 model was found to be too slow, which is consistent with the TM models showing relatively low contributions of northern hemispheric emissions. The anthropogenic contribution varies between 57 and 73%, with inversions accounting for process-specific information through the use of isotopes showing a smaller range of 60–63%.

## 3 Treatment of atmospheric sinks

In this section, we discuss the treatment of atmospheric methane oxidation in inversions. The change in CH$_4$ mixing ratio in an air parcel $i$ due to local sources and sinks is described by

$$\frac{\partial z_i}{\partial t} = E_i - \sum_j k_{i,j}[Ox_j]_i z_i + \sum_l T_l z_l, \tag{1}$$

with $z_i$ and $[Ox_j]_i$ the mixing ratios of CH$_4$ and its main photochemical oxidants OH, Cl, and O($^1$D), reacting at rate $k_{i,j}$. $E_i$ is the emission into air parcel $i$ and transport operator $T_l$ accounts for the advection and mixing of $z_i$ with its surroundings $l$ ($l$ includes $i$). The purpose of inverse modeling is to estimate scaling factors $\mathbf{x}$ of the surface emissions and chemical transformation rates by fitting atmospheric transport model simulated mixing ratios to a set of measurements $\mathbf{y}$. The relation between model simulated measurements $\mathbf{z}^f$ and the sources and sinks of CH$_4$ can be expressed as

$$\mathbf{z}^f = \mathbf{HM}_e\mathbf{x}_e - \mathbf{HM}_s\mathbf{Zx}_s + \mathbf{HM}_0\mathbf{Z}_0\mathbf{x}_0 \tag{2}$$

where $\mathbf{M}$ is a linear chemistry and transport operator translating the state vector $\mathbf{x}$ into model simulated CH$_4$ mixing ratios $\mathbf{z}$, which are sampled using observation operator $\mathbf{H}$ to obtain $\mathbf{z}^f$. The notation follows Rayner et al. (2016) as much as possible, except that we separate the state vector $\mathbf{x}$ in its source, sink, and initial concentration components indicated by subscripts "$e$","$s$" and "$0$". We use $\mathbf{M}$ because the transport model propagates the concentration state, needed to compute the methane sink, from one time step to the next (is an equation 1). Matrix $\mathbf{Z}$ is introduced because de state vector components $\mathbf{x}_s$ and $\mathbf{x}_0$





are usually not defined at the dimension of the modelled mixing ratios $\mathbf{z}$. $\mathbf{Z}$ is defined such that the product $\mathbf{Zx}$ yields $\mathbf{z}$ scaled according to the definition of $\mathbf{x}$. Note that the model simulated observations $\mathbf{z}^f$ are not linearly dependent on $\mathbf{x}_s$, because, unlike $\mathbf{x}_e$, the sink magnitude depends on the methane mixing ratio $\mathbf{z}$, which is influenced by changes in $\mathbf{x}_s$. This introduces a non-linearity in CH$_4$ inversions that optimize the transformation rate. In the Bayesian formulation of the cost function, the *a priori* estimate of $\mathbf{x}_s$ is usually derived from a chemistry transport model (CTM). In that model the oxidant abundances also depend on the mixing ratio of CH$_4$, adding further non-linearity. Following our notation, the CTM changes $[Ox_j]$ in equation 1, which is incorporated in $\mathbf{M}_s$ of equation 2. This means that when photochemical feedbacks are taken into account, $\mathbf{M}_s$ becomes a non-linear operator.

Since CH$_4$ is a long-lived gas, i.e. long compared with the typical time window of inversions, the uncertainties in its sources and sinks influence only a small fraction of its average mixing ratio. Therefore, as long as the inversion uses realistic initial concentrations, e.g. derived from the global surface network, and the *a priori* source and sink estimates are in reasonable balance with the observed global growth rate, the relative changes in $\mathbf{z}$ remain minor. In this case, the inverse problem is only weakly non-linear. As we have seen, the CTM calculated oxidant fields are usually applied to the inversion after correcting global mean OH to match the measurement-inferred lifetime of MCF of 5.5±0.2 yr (Montzka et al., 2011). This step eliminates any modification of global mean OH in the CTM in response to updated CH$_4$ concentrations coming from the inversion. Because of this, the influence of optimized CH$_4$ mixing ratios on CTM calculated OH is usually ignored. Besides global mean OH, it seems reasonable to assume that as long as the relative modifications in CH$_4$ remain small, changes in CTM calculated OH distributions are not significant.

What remains to be accounted for is the non-linearity introduced by optimizing the transformation rate $\mathbf{x}_s$ within the uncertainty of the methyl chloroform analysis. For this purpose, equation 2 can be linearized around an approximation of the CH$_4$ mixing ratio ($\mathbf{z}_n$) as follows,

$$\mathbf{z}_{n+1}^f = \mathbf{HM}_e \mathbf{x}_{e,n} - \mathbf{HM}_s \mathbf{Z}_n \mathbf{x}_{s,n} + \mathbf{HZ}_0 \mathbf{x}_{0,n} \qquad (3)$$

which can then be used to solve the inverse problem using the iterative procedure

$$\mathbf{x}_{n+1} = \mathbf{x}_n - (\mathbf{M}_n^T \mathbf{H}^T \mathbf{R}^{-1} \mathbf{HM}_n + \mathbf{B}^{-1})^{-1} (\mathbf{M}_n^T \mathbf{H}^T \mathbf{R}^{-1} (\mathbf{HM}_n \mathbf{x}_n - \mathbf{y}) + \mathbf{B}^{-1} (\mathbf{x}_n - \mathbf{x}^b)). \qquad (4)$$

Here $\mathbf{x}_n$ is a trial state vector after iteration $n$ (combining "$e$", "$s$" and "$0$"). The other elements in this equation follow the standard notation of Rayner et al. (2016). Usually, the *a priori* CH$_4$ source and sink estimates lead to an atmospheric state that is realistic enough for equation 4 to converge within only a few iterations.

Equation 4 can be simplified further by ignoring uncertainties in $\mathbf{x}_s$ and solving only for surface emissions. In this case the inverse problem becomes linear, and the analytical solution is obtained in a single iteration. If $\mathbf{x}_n^g$ is replaced by $\mathbf{x}^b$ and $\mathbf{x}_{n+1}^g$ by $\mathbf{x}^a$ then equation 4 indeed reduces to the least squares solution of the linear inverse problem (Tarantola, 2005). The reason why this is commonly done is not primarily out of computational convenience, but rather because surface measurements and satellite retrieved total columns provide insufficient information to distinguish between source and sink influences. If sources and sinks are optimised simultaneously, solutions are obtained where source adjustments compensate for sink adjustments





and vice verse. Depending on the freedom of the inversion to adjust the sink, solutions will be obtained that show unrealistic compensating adjustments between sources and sinks.

To deal with this problem, MCF measurements are used to independently constrain the sink, either within the inversion (see for example Bousquet et al. (2006)) or in a separate inversion preceding the $CH_4$ inversion (see for example Bergamaschi

et al. (2009)). Usually, this step only optimizes a climatological global OH sink, i.e. ignoring year-to-year variations and uncertainties in its geographical distribution. Given the importance of the methane sinks, their estimated temporal variations (Montzka et al., 2011) and the associated uncertainties (Holmes et al., 2013; Voulgarakis et al., 2013; Naik et al., 2013), these methods are not satisfactory. This will be true even more in the future, when MCF mixing ratios approach a new steady state to unreported residual sources at concentration levels that will be difficult to measure accurately (Lelieveld et al., 2006). We will

return to this discussion in section 7.

## 4   The use of isotopes

Using measurements of $CH_4$ mixing ratios, only limited information is obtained about source and sink processes. Attempts have been made to use *a priori* information on spatiotemporal emission patterns to optimize the contribution of specific emission classes. If the state vector is defined at a lower resolution than the model, then the *a priori* emission distribution within the

source regions provides some process-specific information (as in Hein et al. (1997)). For inversions that solve at the resolution of the model grid, process-specific flux patterns can be specified only as temporal and spatial correlations in the *a priori* flux error covariance matrix (as in Bergamaschi et al. (2009)), turning this information into a weak constraint. Alternatively, one may just rely on the *a priori* contribution of each process per grid box and partition the inversion optimized flux accordingly. However, for $CH_4$ the *a priori* patterns themselves are rather uncertain and therefore it is questionable whether these methods

allow any useful process-specific information to be gained from the inversion. The hope may be that this situation will improve in the future with improved measurement coverage, for example, from high resolution satellite imagers capable of separating source processes geographically.

Alternatively, isotopic measurements provide truly independent process-specific information. For this purpose, several inversion studies have used measurements of $\delta^{13}C-CH_4$ (Brown, 1993, 1995; Hein et al., 1997; Bergamaschi et al., 2000; Mikaloff

Fletcher et al., 2004b; Bousquet et al., 2006). So far, however, the impact has been limited because of limitations in network coverage, the low single measurement precision, and differences in calibration standards between laboratories (Levin et al., 2012). Furthermore, this approach requires accurate knowledge of the process-specific isotopic fractionation factors, which are not well separated, for example, for different microbial sources such as ruminants, wetlands, and waste treatment, and may also vary strongly for a single source class depending on specific conditions (Zazzeria et al., 2015; Röckmann et al., 2016).

Nevertheless, a rough distinction is possible between the contribution of emissions from microbial sources (wetlands, agriculture, waste processing), energy use (fossil fuel production and consumption), and biomass burning. In addition, measurement techniques are under development with the potential to significantly improve the availability of high quality data in the future (Röckmann et al., 2016; Eyer et al., 2016).





The additional constraints gained by isotopic measurements can be derived starting from the $^{13}$C analogue of equation 3,

$$\mathbf{R}_z\mathbf{z}^f = \mathbf{HM}_e\mathbf{R}_e\mathbf{x}_e - \mathbf{HM}_s\alpha_{12,s}^{13}\mathbf{R}_z\mathbf{Z}\mathbf{x}_s + \mathbf{HM}_0\mathbf{R}_0\mathbf{Z}_0\mathbf{x}_0, \tag{5}$$

with the diagonal matrix $\mathbf{R}$ containing the $^{13}$C/$^{12}$C ratios of CH$_4$. Likewise $\alpha_{12,s}^{13}$ contains the isotopic fractionation of the oxidation reactions in $s$. Note that this equation and those that follow also apply to CH$_3$D. Equation 5 can be reformulated in

$\delta$ notation as follows

$$\delta_z\mathbf{z}^f = \mathbf{HM}_e\delta_e\mathbf{x}_e - \mathbf{HM}_s(\alpha_{12,s}^{13} + \alpha_{12,s}^{13}\delta_z - \mathbf{I})\mathbf{Z}\mathbf{x}_s + \mathbf{HM}_0\delta_0\mathbf{Z}_0\mathbf{x}_0, \tag{6}$$

where $\delta_z$ and $\delta_e$ contain the isotopic delta values of, respectively, atmospheric CH$_4$ and its emissions. For the derivation of equation 6 see appendix A. Different approaches are taken for solving inversions using isotopic measurements, depending on whether sinks strengths and/or $\delta$ values are optimized. If both are optimized, the use of isotopic measurements introduces

additional non-linearity as the observed $\delta$ values are influenced by the product of source strength and fractionation. In Bousquet et al. (2006), the inverse problem is solved by linearizing equation 6 around the first guess state as in equation 3 and iteratively solving the problem as in equation 4. As in CH$_4$ inversions where sinks are optimized, the problem is only weakly non-linear. If the initial state is realistic, subsequent iterations do not modify the solution significantly. As shown in Hein et al. (1997), equation 6 can be further simplified with a few approximations taking out the non-linearity. If sinks and isotopic fractionation

constants are not optimized, including the delta value of the initial condition, then the inversion becomes linear again (see e.g. Mikaloff Fletcher et al. (2004a)).

     The role of the initial condition in inversions using isotopic measurements has received special attention. Tans (1997) and Lassey et al. (2000) demonstrated that $\delta^{13}$C-CH$_4$ takes longer to reach steady state after a perturbation than CH$_4$ itself. The question was raised how long the spin up time of inversions should be to avoid that errors in the assumed initial concentration

field influence the results. If this time is too short, the inversion may fit the data by compensating errors in the initial condition with artificial emission adjustments. It should be noted, however, that the perturbation recovery time for CH$_4$ is also much longer than the spin-up time that is used in inversions using only CH$_4$ data (i.e. without using isotopes). This does not cause problems, as long as the inverse problem is defined such that the initial condition is given sufficient freedom to be optimized itself. The same holds for $^{13}$CH$_4$ in inversions using isotopic measurements.

Second in importance is the representation of initial spatial gradients that take longest to equilibrate, such as the interhemispheric difference and vertical gradients in the stratosphere. However, as long as these gradient components do not contribute to the global burden (i.e. their global integral adds up to zero), the corresponding relaxation times of both CH$_4$ and $\delta^{13}$C-CH$_4$ remain in the order of the corresponding dynamical mixing times. To demonstrate this point, we performed three simulations using a two-box model with the boxes representing the Northern and Southern Hemisphere (see Figure 2). In the

reference simulation the initial condition is in balance with the steady state, and, as expected in this case, nothing changes during the simulation. In a second simulation, the initial concentrations are modified changing the inter-hemispheric gradient, but without changing the global burdens of CH$_4$ and $^{13}$CH$_4$. As can be seen, this simulation recovers at the time scale of the inter-hemispheric exchange (here set to 1 year). Only in the third simulation, where the initial concentrations are perturbed





without conserving global mass, the recovery times become of the order of the $CH_4$ lifetime. Interestingly, in this case the north-south gradient of $CH_4$ still recovers at the time scale of inter-hemispheric mixing, whereas the gradient of $\delta^{13}C$ takes much longer to equilibrate.

From this experiment it follows that long relaxation times, and therefore long spin-up times, can be avoided if the inversion
is capable of recovering the right initial burdens of $CH_4$ and $^{13}CH_4$. Therefore, the inversion should be given sufficient freedom to achieve this, i.e to correct errors in the *a priori* assumed initial global burdens. Additional errors in the global distribution of the initial concentrations call for a spin up time of the order of the longest dynamical mixing time scale, which is the same for $CH_4$ and $\delta^{13}C$-$CH_4$. The required spin up time can be reduced further, for $CH_4$ and $\delta^{13}C$-$CH_4$, by introducing additional degrees of freedom to the initial condition, such as the difference between the Northern and Southern Hemisphere and between
the stratosphere and troposphere, such that these gradients can be optimized from the data also.

## 5   Application to satellite data

The use of satellites in inverse modeling is attractive because of their superior spatial coverage compared with the surface networks. Although a significant step forwards has indeed been made using SCIAMACHY and GOSAT, especially in regions that are poorly covered by the surface network, the coverage is still limited by the need for clear sky conditions to retrieve $XCH_4$.
In addition, the temporal coverage is limited by the revisit time of the satellite. The most useful remote sensing instruments for the quantification of $CH_4$ emissions from space make use of spectral measurements in the Short Wave InfraRed (SWIR) of Earth reflected sun light. Since these photons have traveled the whole atmosphere twice, the measurements are sensitive to $CH_4$ absorption across the full column, down into the planetary boundary layer where the signals of surface emissions are largest (Frankenberg et al., 2005). To obtain sufficient signal puts requirements on the sun angle, which limits the coverage
at high latitudes. Techniques exist to further reduce these coverage limitations, e.g. using active instrumentation (Ehret et al., 2008), an elliptical orbit (Nassar et al., 2014) or a large measurement swath (Landgraf et al., 2016), but these have not been tested out in space yet. Therefore, further improvements in measurement coverage are expected for future missions.

To make efficient use of the growing stream of space borne greenhouse measurements, inversion methods need adaptation. Important steps in this direction have been taken by the application of the 4D-VAR technique to the inversion of $CH_4$ emis-
sions (Meirink et al., 2008b; Rayner et al., 2016), which we refer to as the variational approach to avoid confusion about the applicability of "4D" to the optimization of surface emissions. In this technique, the use of an adjoint model allows evaluation of the cost function gradient at computing time and memory costs that do not scale with the number of measurements, as is the case for the classical matrix inversion technique. However, with the growing information content of the data, a growing number of fluxes can independently be resolved, increasing the required number of iterations. A major limitation of the vari-
ational approach is the use of sequential search algorithms to minimize the cost function. Each step in the sequence involves an evaluation of the cost function gradient, requiring a forward and adjoint model simulation for the full time span of the inversion. Because this procedure is strictly sequential, it is difficult to take advantage of the computational power of modern massive parallel computers. Although parallel search algorithms exist (see e.g. Desroziers and Berre (2012)), they have not





been applied to $CH_4$ emission optimization yet. An alternative approach is the use of ensemble methods such as the ensemble Kalman filter (Peters et al., 2007; Bruhwiler et al., 2014), which allows efficient use of large numbers of processors, although the number of regions for which emissions are estimated are still far less compared with the variational approach.

Finding the solution of a large dimensional inverse problem is not the only challenge in using satellite data. Estimating

the corresponding posterior uncertainties is an even harder computational problem to solve, because methods to approximate the Hessian of the cost function (i.e. the inverse of the posterior covariance matrix) tend to converge slower than the solution itself. This is true in particular at the smallest spatio-temporal scales that are solved for, hence the problem is expected to get worse moving to higher resolutions using instruments that provide more spatio-temporal detail (Meirink et al., 2008b). High-resolution posterior uncertainty estimates are particularly useful in Observing System Simulation Experiments (OSSEs) for

testing the performance of inversions using new concepts for measuring greenhouse gases from space (Houweling et al., 2005; Miller et al., 2007; Hungershoefer et al., 2010). A popular method to derive such uncertainties is a Monte Carlo application of the variational approach introduced by Chevallier et al. (2007). This method is computational demanding, however, because of the large number of inversions needed to determine the posterior uncertainty at a precision of a few % (Pandey et al., 2015). More precise methods exists (Rödenbeck, 2005; Hungershoefer et al., 2010), but they can only be applied to the uncertainty of

a limited number of fluxes.

To sample the model for comparison to satellite retrievals involves application of the retrieval averaging kernel to the modeled vertical profile of $CH_4$ as follows,

$$\mathbf{z}^f = \mathbf{t}_l^T \mathbf{A}_{l,l} \mathbf{z}_l + \mathbf{t}_l^T (\mathbf{I}_{l,l} - \mathbf{A}_{l,l}) \mathbf{z}_l^b \qquad (7)$$

here the total column operator $\mathbf{t}_l$ contains normalized partial columns $\Delta p_i / p_{surf}$ for each layer $i$ of the retrieved profile of $l$

layers. The product of $\mathbf{t}_l^T$ and the profile averaging kernel $\mathbf{A}_{l,l}$ is the column averaging kernel $\mathbf{a}_l$. If the retrieval uses profile scaling then only $\mathbf{a}_l$ is available (see e.g. Borsdorff et al. (2013)). $\mathbf{z}_l$ is the modeled vertical profile of methane at the vertical grid of the retrieval. Since the averaging kernel values depend on the vertical discretization of the retrieved state, the model profile should be discretized the same way (i.e according to the retrieval grid). Furthermore, the averaging kernel depends on the unit in which the state vector (of the retrieval) is expressed, either absorber amount or mixing ratio, and so the model

profile has to be expressed accordingly (Deeter et al., 2007). It is advised to correct errors in the $CH_4$ column amount due to regridding from the vertical grid of the model to that of the retrieval to ensure that the regridding conserves mass. $\mathbf{z}_l^b$ is the *a priori* profile that was used in the retrieval, and $\mathbf{I}_{l,l}$ is the identity matrix. For retrievals that use profile scaling the second RHS term in equation 7 should be close to zero (see Borsdorff et al. (2013)). Deviations point to the use of a different *a priori* profile than was used in the retrieval.

For proxy $XCH_4$ retrievals the use of equation 7 introduces an additional complication, because of the way information about $CH_4$ and $CO_2$ is combined. The correct way to deal with this can readily be understood looking at its equation

$$XCH_4^{proxy} = \frac{XCH_4^{ret}}{XCO_2^{ret}} . XCO_2^{mod}, \qquad (8)$$

showing how the proxy retrieval is derived from the ratio of non-scattering retrievals $XCH_4^{ret}$ and $XCO_2^{ret}$, and a model-derived estimate $XCO_2^{mod}$ as discussed already in section 2. Suppose that $XCO_2^{mod}$ and $XCO_2^{ret}$ were perfect, then the





contribution of $CO_2$ to the RHS of equation 8 would cancel out. However, this also requires that the averaging kernel of $XCO_2^{ret}$ is applied to $XCO_2^{mod}$, which is therefore the correct way to specify $XCO_2^{mod}$. What remains is weighted according to the averaging kernel of $XCH_4^{ret}$, which is the one that should be used when applying equation 7 to proxy $XCH_4$ retrievals. In a ratio inversion, equation 7 is applied separately to the modeled profiles of $CO_2$ and $CH_4$, after which the ratio of the

modeled total columns is taken.

    The use of averaging kernels could in theory be avoided by including the radiative transfer model in the inversion, so that the model yields the satellite observed spectral radiances instead of retrieved mixing ratios. However, for practical reasons this has not been done so far. This approach would avoid inconsistencies between the *a priori* profile and its uncertainty as used in the retrieval and as generated by the *a priori* transport model. For further discussion about the statistical consequences of this

inconsistency see Chevallier (2015).

    A major challenge in the use of satellite data in inversions is to realistically account for uncertainty. Satellite retrievals are influenced by various physical and chemical conditions along the light path that is being measured. Inaccuracies in the capability of the retrieval to take these into account vary at the same spatiotemporal scales as these conditions themselves. They may even correlate with the retrieved variable, like water vapor in the case of SCIAMACHY $XCH_4$ retrievals (Frankenberg

et al., 2008; Houweling et al., 2014), which makes it difficult to distinguish signal from error. Errors that behave quasi-random and affect neighboring retrievals in a coherent way, can in theory by accounted for by specifying the off-diagonal terms in the data error covariance matrix. In practice, there are many ways to do this, but quantitative information is lacking to justify a specific choice. In general, correlated uncertainty reduces the number of independent measurements, which justifies the averaging of retrievals within a certain distance of each other. Usually the uncertainty of the mean is calculated using a lower

bound representing the contribution of purely systematic error. An alternative approach, referred to as "error inflation", is to increase the error of individually assimilated retrievals such that the uncertainty of a mean of surrounding retrievals does not drop below this minimum level (Chevallier, 2007). The advantage of this approach is that it avoids subjective decisions about which samples to combine into an average. Error inflation, or similar methods that compensate the neglect of off diagonals in the data error covariance matrix by increasing the (diagonal) uncertainty, lead to a $\chi^2$ below 1. Although this may seem

sub-optimal from a statistical point of view, Chevallier (2007) demonstrated that this de-weighing of data nevertheless leads to uncertainty reductions that are closer to those obtained when off-diagonals in **R** had been accounted for. Therefore, this approach avoids over-constraining the problem by neglecting the contribution of data error covariance.

    As the inversion formalism assumes all errors to be random, measurement bias must either be corrected prior to use in the inversion, or be estimated as state vector elements. Both cases require knowledge of the spatiotemporal pattern of the bias.

Given a model representation of the bias $\mathbf{H}'$, the model simulated measurements can be reformulated as,

$$\mathbf{z}^f = \mathbf{HMx} + \mathbf{H}'(\mathbf{x}'), \tag{9}$$

where systematic errors are accounted for using a set of extended state vector elements $\mathbf{x}'$. Different formulations of $\mathbf{H}'(\mathbf{x}')$ have been used in $CH_4$ inversions using satellite retrievals from GOSAT and SCIAMACHY. Bergamaschi et al. (2007) use simple polynomials of latitude and season, to account for inconsistencies arising from the combined use of surface and satellite




measurements at large scales. The motivation is that surface measurements are best suited to constrain the large scales, whereas satellite data can be used to fill in regional detail, which the surface network is unable to resolve. An alternative approach (Houweling et al., 2014) is to assess potential causes of systematic error in satellite retrievals, identify the main drivers - or variables that can serve as proxies of their spatiotemporal variation - and optimize the magnitude of the corresponding error

contribution in the inversion. It should be noted that the inversion optimized $\mathbf{x}'^a$ has contributions from the measurements as well as systematic errors in the transport model. In addition, if the bias variables covary with the $XCH_4$ signal of uncertainties in the surface fluxes, then the inversion will have limited skill in resolving their contributions. To avoid this problem, the TCCON network can be used to optimize the bias model (Wunch et al., 2011b; Houweling et al., 2014). However, because of its sparse global coverage and uncertainties in the TCCON measurements themselves, uncertainties will remain that can be

further optimized in the inversion.

## 6   The importance of transport model uncertainties

An important assumption in inverse modeling is that the influence of atmospheric transport model uncertainties is small compared with the uncertainty of the *a priori* fluxes. Formally, there are ways to account for transport model uncertainty in the optimization, however, in practice they are difficult to implement lacking the information required to characterize the statistics

of transport model uncertainties in a realistic manner. In addition, there is the fundamental problem that the transport model uncertainty has a significant and poorly quantified systematic component. Patra et al. (2011) assess the importance of transport model uncertainties, based on the results of the TransCom-$CH_4$ model inter-comparison. The results highlight the importance of specific aspects of atmospheric transport that are critical for the simulation of atmospheric methane, as will be discussed further in this section. To quantify the impact of transport model differences on inversion-estimated surface fluxes requires an

inversion inter-comparison. Attempts in this direction have been made, for example by Kirschke et al. (2013), however, in that study inversions have been compared without a protocol to standardize the setups. Although useful for an assessment of global $CH_4$ emissions and uncertainties, to isolate the role of transport model uncertainties requires a dedicated experiment. Locatelli et al. (2013) used the output of 10 models participating in the TransCom-$CH_4$ experiment to generate 'pseudo measurements' that were inverted in the LMDz model. The results confirm the importance of transport model uncertainties, with estimated

annual fluxes on sub-continental scales varying by 23–48%.

Several studies have highlighted the large contribution of atmospheric transport, including intra and inter-annual variability, to the observed variability in $CH_4$ (Warwick et al., 2002; Patra et al., 2009; Terao et al., 2011). Because of this, studies which directly relate mixing ratio variations to source variations should be treated with care (see e.g. Bloom et al. (2010) and Turner et al. (2016)). For $CH_4$ inversions the observed inter-hemispheric gradient is of particular importance as it is the dominant

mode of variation in background $CH_4$ mixing ratios. The results of Patra et al. (2011) point to a ~50 ppb (40%) difference in the simulation of this gradient due to differences in model simulated inter-hemispheric exchange times. Uncertainty in inter-hemispheric exchange not only affects the inversion-derived latitudinal distribution of emissions, but also its seasonal cycle in the Tropics. The latter is caused by the seasonal dynamics of the intertropical convergence zone (ITCZ). The observed seasonal





cycles at tropical measurement sites, such as Samoa and Seychelles, are largely determined by the seasonally varying position of the ITCZ in combination with the size of the north-south gradient of $CH_4$. To assess and improve the inter-hemispheric exchange in models, $SF_6$ measurements have been used (Patra et al., 2011; Monteil et al., 2013). Despite sizable uncertainties in the emission inventory of $SF_6$ (Levin et al., 2010), it nevertheless provides an important constraint on inter-hemispheric

exchange.

Large $CH_4$ gradients are found also in the stratosphere, owing to the long time scale of stratosphere-troposphere exchange in combination with the chemical degradation of $CH_4$ in the stratosphere. The modeling of stratospheric $CH_4$ is gaining importance with the increasing use of satellite data in source–sink inversions. The offline atmospheric transport models that are used for inverse modeling tend to underestimate the residence time (or "age") of stratospheric air (Douglass et al., 2003;

Bregman et al., 2006). As a consequence of this, models which accurately reproduce the surface concentrations as observed by the global networks (e.g. after optimization using those data) are expected to overestimate satellite observed total column $CH_4$ (Locatelli et al., 2015). Since the mean age of stratospheric air varies latitudinally and seasonally, the transport bias varies accordingly. Indeed, the TransCom-$CH_4$ simulations (Patra et al., 2011) show large differences between models increasing towards higher latitudes in the stratosphere. Although the averaging kernel of SWIR $XCH_4$ retrievals decreases with altitude in

the stratosphere, transport model differences are large enough to be important for emission quantification. Satellite instruments capable of measuring stratospheric $CH_4$, such as MIPAS and ACE-FTS, are useful for testing models. However, the accuracy of those measurements is also limited (Ostler et al., 2016). A promising development is the use of air core to measure the stratospheric profile of $CH_4$ at high accuracy (Karion et al., 2010), not only to evaluate the accuracy of total column FTS measurements from the TTCON network but also atmospheric transport models. However, because of the observed local

variability, which coarse grid models have difficulty reproducing, many balloon flights will be needed to assess and improve, or bias correct, the models.

Using continuous measurements from dense regional networks within Europe and the USA the large-scale transport problems discussed above are less important. In this case, the observed variability is determined mostly by the passage of fronts of synoptic weather systems and the planetary boundary layer (PBL) dynamics. Although the emissions of methane from energy

use have some diurnal variation, the PBL dynamics are more important especially during summer. Unfortunately, the representation of the nocturnal boundary layer in transport models is too poor to make use of the observed diurnal variability. Instead, measurements are used only during the afternoon when the planetary boundary layer is well developed. Nevertheless, the mixing within the PBL and the trace gas exchange with the free troposphere is an important source of uncertainty (Kretschmer et al., 2012; Koffi et al., 2016). Since satellite data from sensors operating in the SWIR are only weakly sensitive to the vertical

distribution of $CH_4$ in the troposphere, their use may be less sensitive to such errors. The increased coverage and spatial resolution of the new generation of satellite sensors, such as Sentinel 5 precursor TROPOMI (Landgraf et al., 2016), will increase the relevance of satellites for regional-scale emission assessment. These data are highly complementary to surface measurements in the sense that they have a different sensitivity to critical aspects of transport model uncertainty. To bring together regional emission estimates from satellites and surface data is both a major challenge, and a great opportunity for testing atmospheric

transport.





To assess transport model uncertainties in the simulation of $XCH_4$ we analyzed the archived output of the TransCom-$CH_4$ experiment (Patra et al., 2011) (see Figure 3). $XCH_4$ fields were calculated from monthly mean mixing ratio output on pressure levels for the year 2000, interpolated to a common horizontal resolution of $2°x2°$. To account for the vertical sensitivity of satellite retrieved $XCH_4$, we apply averaging kernels from the RemoTeC GOSAT full physics retrieval (Butz et al., 2011).

Finally, standard deviations were calculated for each vertical column using results from 7 models: ACTM, GEOS-CHEM, MOZART, NIES, PCTM, TM5, and TOMCAT. Figure 3 shows 1-$\sigma$ differences between the models for the total column, as well as the percentage contribution of stratospheric and tropospheric sub-columns. Results for the total column show $\sigma$ values up to $\sim2\%$ (or 35 ppb), associated mostly with steep orography, see e.g. the Andes, the Himalayas, and most notably the ice caps. The contribution from the troposphere is low in the Southern Hemisphere compared with the Northern Hemisphere, because a global

offset between the models has been removed at the South Pole. Therefore, the impact of differences in inter-hemispheric mixing is seen mostly in the Arctic, contributing $\sim10$ ppb in $XCH_4$. The tropospheric contribution to transport model uncertainty also highlights the centers of tropical convection. The contribution of the stratospheric column to the variation in $XCH_4$ is sizable, and increases towards the poles. The asymmetry between north and south pole is mainly because of the South Pole correction, taking out offsets in the SH lower troposphere. It means that the large uncertainties in $XCH_4$ over Antarctica are caused

mainly by the stratosphere. They follow the orography of the ice cap, because the impact of the stratospheric sub-column increases as the thickness of the tropospheric sub-column reduces. Towards northern latitudes differences increase up to 50–60%, highlighting the importance of uncertainty in stratospheric transport when inverting satellite retrieved total columns.

## 7  New directions

Compared with the 1990s, when the first inverse modeling studies on $CH_4$ were published, many things have changed, most

notably the availability of data and computer power. Inverse modeling techniques have been developing further to make use of these advances. Studies concentrated on the use of specific data sets, for example, to investigate the use of remote sensing, or tall tower networks. Other types of measurements were used to further constrain specific processes, such as the use of MCF to constrain OH or satellite-observed inundation to improve the representation of wetland dynamics. Despite these efforts, the robustness of inverse modeling derived estimates is still limited for scales smaller than broad latitudinal bands (Kirschke

et al., 2013). Because of this, it remains difficult to attribute the significant changes in the global growth rate that have been observed in the past decades to specific processes. Besides important efforts to further improve the quality of data and models, there is scope to further explore the combined use of different datasets to further constrain the inverse problem from different directions.

To improve our understanding of what drives the inter-annual variability in the $CH_4$ growth rate it is critical to be able to

separate influences from varying sources and sinks. To this end, further effort is needed to constrain the atmospheric oxidation of $CH_4$. Since the $CH_4$ and $\delta^{13}$C-$CH_4$ data provide limited constraints on the sink, other data will be needed. The problem with the MCF optimization method is that the two measurement networks, NOAA and AGAGE, lead to different answers, which again differ from chemistry transport model simulations, as demonstrated nicely by Holmes et al. (2013). A better understand-



ing is needed of what causes these differences, including the role of the sparse network for measuring MCF and remaining questions regarding radical recycling in CTMs (Lelieveld et al., 2016). A promising direction is the use of measurements of other key compounds in photochemistry, such as CO, $O_3$, $CH_2O$, and $NO_x$ to bring the photochemical models in better agreement with the actual observed state (Miyazaki et al., 2012).

Measurements of the vertical profile of $CH_4$ may further improve the separation between surface sources and atmospheric sinks. For this purpose, aircraft measurements are available, as well as satellites that are particularly sensitive to specific altitudes, such as IASI (Cressot et al., 2014) and TES (Worden et al., 2012). However, since the gradients in the free troposphere are small, and influenced by uncertainties in the sub grid parameterization of vertical transport in transport models, it is questionable how effective aircraft profile measurements can be. In the stratosphere the prospects for independent measurement

constraints on the sinks are better, since the gradients are much larger. It will still require the combined use of $CH_4$ and a chemically inert tracer such as $SF_6$ or $CO_2$ to distinguish between uncertainties in stratospheric transport and chemistry.

   Another approach to separate the influences of sources and sinks is to limit the domain of the inversion to important source regions. In such regions, the concentration signal of the sources is much more variable than that of the sink because of the high spatial heterogeneity of $CH_4$ emissions. The sink scales with the total $CH_4$ abundance, which is still dominated by the

background. The sources can be quantified, independent of the sink, using short-term departures from the background due to fresh emissions. The influence of the sink on those departures can be neglected, because the regional transport times are much shorter than the lifetime of $CH_4$. Meanwhile, several successful attempts have been made to quantify regional $CH_4$ emissions using this approach, for example, using the tall tower networks in Europe (Bergamaschi et al., 2010, 2015), and the U. S. A. (Miller et al., 2013; Bruhwiler et al., 2014). Continuous measurements from tall towers record highly variable $CH_4$

concentrations providing much information about regional sources, challenging the performance of high-resolution mesoscale transport models. Despite the challenges, the results demonstrate its potential for supporting country scale emission verification.

   The use of land surface models to provide *a priori* emission estimates for use in inverse modeling implies that the concept of carbon cycle data assimilation (CCDAS), which has only been applied to $CO_2$ so far, may be beneficial for $CH_4$ also. Besides the advantage of gaining actual process understanding, which is needed for improved projections of future $CH_4$ concentrations,

optimization at process level facilitates the combined use of different types of measurements. In the case of wetland emissions, hydrological conditions are an important driver, particularly in the Tropics (Ringeval et al., 2014). Satellite observed inundation is already used to prescribe the dynamics of the wetland extent (Ringeval et al., 2010; Pison et al., 2013). In combination with hydrological modeling, some limitations of the measurements could be addressed, such as the difficulty to measure water underneath dense vegetation and the fact that wetland soils may be partially saturated but are not necessarily inundated.

To improve the representation of wetland emissions in process models will also require extension of the flux measurement network. These measurements would be an essential component of a multi-stream data assimilation system for methane (or MDAS), but the coverage of the network should be more comparable to that of FLUXNET $CO_2$ (Baldocchi et al., 2001). In particular, the model parameterization of methane emissions from tropical wetlands is severely limited by the availability of flux measurements. Such limitations are important to address, but in the meantime the concepts and methods for MDAS should

already be developed using the existing data.



Besides the use of satellites to improve the representation wetland hydrology, several other kinds of measurements can provide process-specific information. For example, atmospheric tracers such as ethane and carbon monoxide provide useful information about emissions from fossil fuel mining (Simpson et al., 2012; Aydin et al., 2011; Hausmann et al., 2016) and biomass burning (Wilson et al., 2016; Bastos et al., 1995), which could be combined with methane measurements in a data

assimilation framework. Figure 4 shows a conceptual diagram of how current inversion setups could evolve in order to further increase the constraints on the source and sink processes from the use of of various types of measurements.

## 8   Closing remarks

In the past three decades of $CH_4$ inverse modeling important progress has been made in developing atmospheric transport models and inversion methods for the use of various kinds of measurements. Despite this progress, it remains a challenge to

identify the dominant drivers of the large global growth rate variations that have been observed during this period. This is caused in part by the difficulty of separating the influence of surface emissions and atmospheric sinks. Breaking up global estimates into regional contributions, the robustness of the estimates decreases further, except in regions where tall tower networks support regional flux estimation. There is no single solution to this problem since every new approach, such as the use of methane isotopologues or satellite data, brings new information as well as additional unknowns. To make optimal use

of the improving observational constraints on atmospheric methane puts increasing demands on the quality of atmospheric transport models. We demonstrated that the use of satellite retrieved $XCH_4$ calls from an improved model representation of stratospheric methane.

Besides the ongoing developments to improve models and measurement datasets, the combined use of different datasets in a single optimization framework is still left largely unexplored. As discussed, the methane budget offers several directions

for applying the CCDAS concept to $CH_4$, on the side of the sources, the sinks, or, ideally both. It will be a challenge to combine different datasets in a consistent manner, but inconsistencies will also help identify new directions for improvement. The use of isotopic measurements was discussed and how the initial condition can be set up to avoid influences of long isotopic equilibration times.

The COP21 climate agreement offers a great opportunity for inverse modeling to support international efforts to reduce

emissions, by providing independent estimates to verify if intended reduction targets are being achieved. However, the steps that are needed to become relevant in this process are still sizable. Compared with the achievements of the past three decades, it is clear that the overall progress will have to accelerate. To achieve this will require a closer international collaboration, to make more efficient use of the collective effort that is spend by different research groups already. The annual assessments of GCP-CH4 (Saunois et al., 2016) are an important first step in this direction.





## Appendix A: Isotopic equation in delta notation

To derive equation 6 from equation 5 we first subtract equation 2, multiplied with a reference isotopic ration $\mathbf{R}_{ref}$, from equation 5 resulting in

$$(\mathbf{R}_z - \mathbf{R}_{ref})\mathbf{z}^f = \mathbf{HM}_e(\mathbf{R}_e - \mathbf{R}_{ref})\mathbf{x}_e - \mathbf{HM}(\alpha^{13}_{12,s}\mathbf{R}_z - \mathbf{R}_{ref})\mathbf{Z}\mathbf{x}_s + \mathbf{HM}(\mathbf{R_0} - \mathbf{R}_{ref})\mathbf{Z}_0\mathbf{x}_0, \tag{A1}$$

5  to transfer to delta notation we substitute $\mathbf{R}$ in equation A1 using

$$\mathbf{R}_x = (\mathbf{I} + \delta_x)\mathbf{R}_{VPDB} \tag{A2}$$

where subscript $x$ refers to any specific occurrence of $\mathbf{R}$ in equation A1 and $\mathbf{R}_{VPDB}$ is the isotopic ratio of the Vienna Pee Dee Belemnite international reference standard. Note that since we are using matrix notation $\mathbf{R}$ represents a diagonal matrix of isotopic ratios (this same applies to $\delta$ and $\alpha$). After substitution and dividing by $\mathbf{R}_{VPDB}$ we obtain

10  $$(\delta_z - \delta_{ref})\mathbf{z}^f = \mathbf{HM}(\delta_e - \delta_{ref})\mathbf{x}_e - \mathbf{HM}(\alpha^{13}_{12,s}\delta_z + \alpha^{13}_{12,s} - (\delta_{ref} + \mathbf{I}))\mathbf{Z}\mathbf{x}_s + \mathbf{HM}(\delta_0 - \delta_{ref})\mathbf{Z}_0\mathbf{x}_0. \tag{A3}$$

Depending on the choice of $\delta_{ref}$ different formulations can be derived. Equation 6 is derived using $\delta_{ref} = 0$.

*Acknowledgements.* We acknowledge the support from the International Space Science Institute (ISSI). This publication is an outcome of the ISSI's Working Group on "Carbon Cycle Data Assimilation: How to consistently assimilate multiple data streams".



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



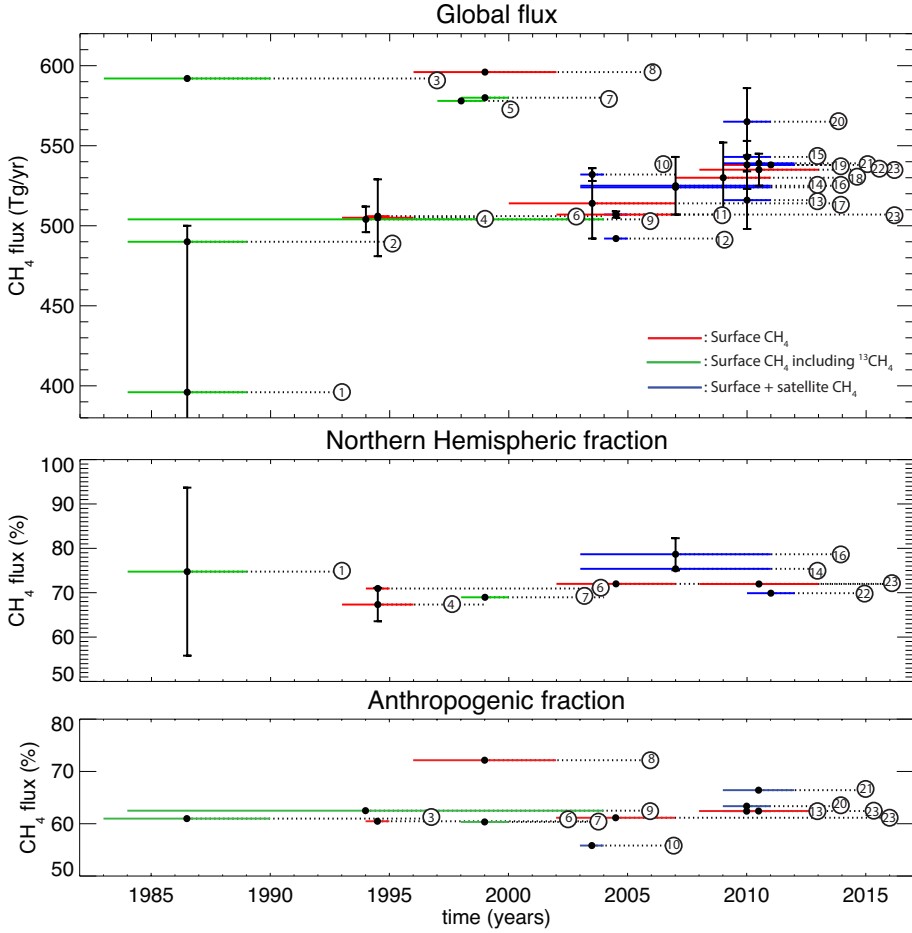

**Figure 1.** Evolution of inversion-derived estimates for the global total CH$_4$ flux (top), its hemispheric distribution (middle), and the anthropogenic contribution (bottom). Horizontal solid lines indicated the time range of the estimate. The right end of dotted lines point to the date of publication. Note that the CH$_4$ trends that are seen are influenced by the evolution of the inversion methods that were used. Numbered circles refer to publication references, as follows: 1:Brown (1993); 2:Brown (1995); 3:Hein et al. (1997), inv.S0; 4:Houweling et al. (1999); 5:Bergamaschi et al. (2000); 6:Wang et al. (2004); 7:Mikaloff Fletcher et al. (2004b), inv.S2; 8:Chen et al. (2006); 9:Bousquet et al. (2006); 10:Bergamaschi et al. (2007), inv.S3; 11:Bergamaschi et al. (2009), inv.S1; 12:Pison et al. (2009); 13:Fraser et al. (2013), inv.3; 14:Bergamaschi et al. (2013), inv.S1SCIA; 15:Monteil et al. (2013), inv.FPNO; 16:Houweling et al. (2014), inv.SQflex; 17:Bruhwiler et al. (2014); 18:Bruhwiler et al. (2014); 19:Cressot et al. (2014); 20:Cressot et al. (2014); 21:Turner et al. (2015); 22:Alexe et al. (2015); 23:Patra et al. (2016).





**Figure 2.** Box model calculated relaxation times to hemispheric disturbances in $CH_4$ and $^{13}CH_4$ with respect to a steady state equilibrium ("Eq"). Theoretical disturbances of the steady state are either global mass conserved ("MC") or not ("D.Eq").







**Figure 3.** Uncertainty in XCH$_4$ due to transport model differences for January (top) and July (bottom). The middle and right panels show the percent contribution from the troposphere (1000–200 hPa) and the stratosphere (200–0 hPa) to the total column variability shown in the left panel. Results are obtained using the submissions to the TransCom-CH$_4$ experiment (Patra et al., 2011) (CTL tracer for the year 2000).





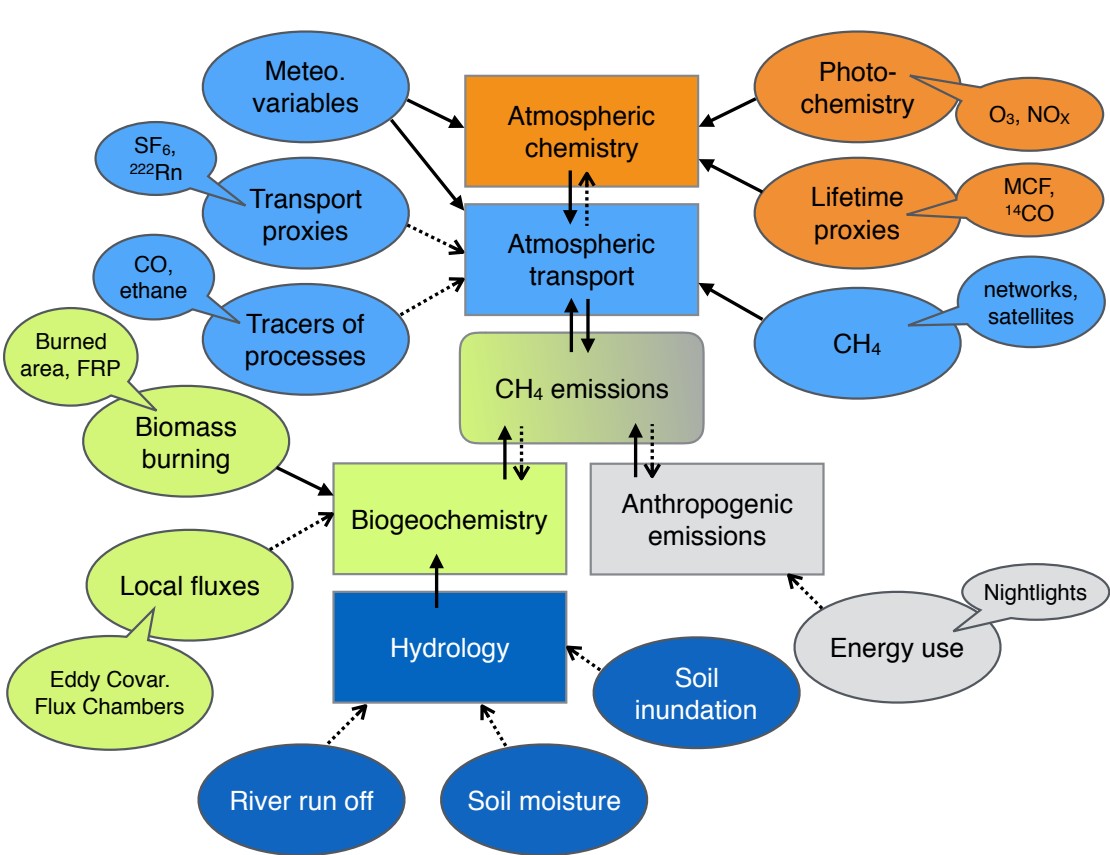

**Figure 4.** Conceptual diagram of ways to extend the use of measurements in CH$_4$ flux inversions. Square boxes represent models, ovals measurements, and the rounded box represents the target variable of the CH$_4$ inversion. Call outs provide examples of the kind of measurements that are meant by the ovals (without attempting to be complete). Black arrows: Coupled / assimilated in inversions already; Dashed arrows: Not (yet) coupled / assimilated.