# Peer review of "Global inverse modeling of CH4 sources and sinks: An overview of methods"

_Atmospheric Chemistry and Physics, 2016_

## Referee Comment (RC1) · Anonymous Referee #1 · 5 Aug 2016

General comments

Paper reviews the current state of methane flux inverse modeling. The historical prospective is also presented. The paper is well written and can be published after rather minor revisions.

Detailed comments.

Page 10 Line 11. Authors refer to Monte Carlo application of the variational approach as a method of choice for uncertainty estimates and note that it is computationally demanding. It should be mentioned that Meirink et al (2008b), see Eq. 8, presented an analytical method for uncertainty estimates, that uses singular vectors retrieved during a single run of iterative optimization process, instead of multiple runs required in randomization approach.

[Figure]

Page 11 Line 6. It is difficult to understand how the use of radiative transfer model in inversion in place of using retrieved profile and averaging kernel matrix would make analysis simpler. The problem of altitude dependence of observed signal, which is different between carbon dioxide and methane, is not going away after incorporating retrieval process in inversion.

Page 15, Line 5. Authors write: "Measurements of the vertical profile of CH4 may further improve the separation between surface sources and atmospheric sinks." This appears as overstatement. As authors admit in the same paragraph, the OH sink-related gradients in troposphere are too small to measure.

---

## Referee Comment (RC2) · Anonymous Referee #2 · 17 Aug 2016

Overview:

The manuscript "Global inverse modeling of CH4 sources and sinks: An overview of methods" by Houweling et al. provides an analysis of the current status of the application of inverse modeling techniques to methane flux estimation, along with a discussion of both their history and their future potential. This paper is a useful documentation and provides some clear ideas for the future. I recommend publication after a few minor changes.

Section 3: Perhaps some mention (here or elsewhere) of the fact that the tropospheric chlorine sink of methane is often not included in inversions. Although relatively small, this will have had some effect on the inversion results that you show in Figure 1.

Page 13, lines 3-5: The authors write that SF6 "provides an important constraint on

inter-hemispheric exchange". This should be explained a little further. Do the authors mean that the SF6 observations should be used actively within an inversion in some way, in order to contain the inter-hemispheric transport? Or as in Monteil et al., (2013) in order to improve the advection parameterisation before an inversion is undertaken?

Page 15, line 5: The authors state that: "Measurements of the vertical profile of CH4 may further improve the separation between surface sources and atmospheric sinks." This statement should be expanded upon, as it is not clear how this would be true given the long lifetime of methane.

Technical corrections:

Page 16, line 14: "To make" -> "making"

Page 16, line 16: "from" -> "for"

Figure 3: The use of a blue-red colorbar in this figure implies positive and negative contributions. You should consider changing to non-diverging colors.
* * *

---

## Referee Comment (RC3) · Anonymous Referee #3 · 17 Aug 2016

General comments:

The paper presents a comprehensive overview of inverse modeling of methane sources and sinks, describing the state-of-the-art of research in this field, including the 'historical' development, and giving future perspectives. The manuscript will also serve as an instructive introduction to the field. The paper is very well written and is recommended for publication after only few minor revisions.

Specific comments:

Page 4, line 27: CH4 is not absent in the stratosphere, although concentrations are low and a steep vertical gradient exists. Are you referring to CH4 sources, which are absent in the stratosphere?

[Figure]

Although the authors refer to Rayner et al. (2016) for the notation of variables and terms of the equations, it would be helpful for the reader (and better suit the didactical purpose of the paper) to include the definitions directly in the paper.

Page 8, line 1: Isn't this the 13C analogue of the more general equation 2.

Page 17, eq. A1, A3: For consistency subscripts should be used for M.

Technical comments:

Page 5, line 31: ...(as in equation 1). ...the state vector components...

Page 16, line1: ... representation of wetland hydrology,...

Page 16, line 16: from -> for

Page 18, line 14: Röckmann

Page 19, line 18: Please check editor name.

Page 22, line 30: 2. Inverse modeling of CH4 fluxes...

Page 23, line 24-27: Please check spelling of co-author names

Figure 1: It could be helpful to indicate in the graphs whether isotopic information was used. Please specify for references 19 and 20 the scenario that was used in this graph.

Figure 3: ... January (left) and July (right). The middle and bottom panel show ... top panel.

Figure 3: Better not use a divergent color scale because in particular a blue-white-red color scale suggests negative and positive values or deviations from a central value.

---

## Author Comment (AC1) · 11 Oct 2016

**Response to Anonymous Referee 1**

We would to thank the referee for the time and effort spent to help improve our manuscript. The structure of this document is as follows: Referee text's are in Italic font, answers are in roman, modifications to the text are in bold font.

*General comments Paper reviews the current state of methane flux inverse modeling. The historical prospective is also presented. The paper is well written and can be published after rather minor revisions.*

*Detailed comments. Page 10 Line 11. Authors refer to Monte Carlo application of the variational approach as a method of choice for uncertainty estimates and note that it is*

*computationally demanding. It should be mentioned that Meirink et al (2008b), see Eq. 8, presented an analytical method for uncertainty estimates, that uses singular vectors retrieved during a single run of iterative optimization process, instead of multiple runs required in randomization approach.*

This method that the referee refers to is mentioned a few sentences earlier (line 5), where we call it 'methods to approximate the Hessian of the cost function'. As explained in the text, they are problematic for OSSEs, which require uncertainty estimates at the resolution of the model grid. As mentioned in Meirink et al (2008b) the convergence of the method is scale dependent. In our experience it doesn't really work at the grid scale, which is why we mention alternative methods including the Monte Carlo method.

To make a clearer link between the method of Meirink et al (2008b) and 'methods to approximate the Hessian of the cost function' we added a reference as follows: '(i.e. the inverse of the posterior covariance matrix, **see Meirink et al, 2008b for details**)'

*Page 11 Line 6. It is difficult to understand how the use of radiative transfer model in inversion in place of using retrieved profile and averaging kernel matrix would make analysis simpler. The problem of altitude dependence of observed signal, which is different between carbon dioxide and methane, is not going away after incorporating retrieval process in inversion.*

The problem that is addressed here doesn't concern the altitude dependence of the signal, which indeed doesn't change, but rather the inconsistent use of a priori profiles in the retrieval and the sampling of the chemistry transport model. As discussed in Chevallier et al (2015) satellite retrievals make use of a priori constraints, which are much looser than justified by the transport model. In the coupled approach, there is only 1 a priori profile; the one that corresponds to the a priori transport model. Hence this inconsistency does not exist anymore.

*Page 15, Line 5. Authors write: "Measurements of the vertical profile of CH4 may further improve the separation between surface sources and atmospheric sinks." This*

*appears as overstatement. As authors admit in the same paragraph, the OH sink-related gradients in troposphere are too small to measure.*

This caveat in line 7, which the referee mentions here, is given to avoid what the referee is worried about; overstating the potential of vertical gradient measurements. Therefore we do not agree that we are overstating, because we do already what the referee expects us to do. Nevertheless, we'd like to mention the use of vertical profile information, because it should be investigated before we conclude that no useful constraints on OH can be derived. We do not agree that sink related gradients are too small to measure (see our answer to referee 2).
* * *

---

## Author Comment (AC2) · 11 Oct 2016

**Response to Anonymous Referee 2**

We would to thank the referee for the time and effort spent to help improve our manuscript. The structure of this document is as follows: Referee text's are in Italic font, answers are in Roman, modifications to the text are in bold font.

*Overview: The manuscript "Global inverse modeling of CH4 sources and sinks: An overview of methods" by Houweling et al. provides an analysis of the current status of the application of inverse modeling techniques to methane flux estimation, along with a discussion of both their history and their future potential. This paper is a useful documentation and provides some clear ideas for the future. I recommend publication after a few minor changes.*

[Figure]

*Section 3: Perhaps some mention (here or elsewhere) of the fact that the tropospheric chlorine sink of methane is often not included in inversions. Although relatively small, this will have had some effect on the inversion results that you show in Figure 1.*

The following sentence has been added to section 2. Page 5, line 10: **Furthermore, the use of MCF to constrain tropospheric methane oxidation does not account for the contribution of other potentially important oxidants, such as chlorine radicals in the marine boundary layer (Allen et al, 2005).**

*Page 13, lines 3-5: The authors write that SF6 "provides an important constraint on inter-hemispheric exchange". This should be explained a little further. Do the authors mean that the SF6 observations should be used actively within an inversion in some way, in order to contain the inter-hemispheric transport? Or as in Monteil et al., (2013) in order to improve the advection parameterisation before an inversion is undertaken?*

The following sentence has been added to section 6: Page 13, line 3: **So far, transport and methane fluxes have been optimized in separate steps, although they could in theory be combined in a single inversion.**

*Page 15, line 5: The authors state that: "Measurements of the vertical profile of CH4 may further improve the separation between surface sources and atmospheric sinks." This statement should be expanded upon, as it is not clear how this would be true given the long lifetime of methane.*

This is a good point, which we indeed didn't give sufficient thought. To quantify the sensitivity we did two forward runs: 1) The standard TM5 CH4 setup, 2) As setup 1) with sources and sinks increased by 10%. Results of these simulations are evaluated in the 4th year. They represent 2 solutions that yield approximately the same global burden. The question is whether vertical profile measurements could detect the difference. Figure 1 shows those differences averaged seasonally. As can be seen there is an approx. 10 ppb difference between the surface and the tropopause. It is stronger in winter than in summer (when convection reduces vertical gradients). Gradients develop despite the short time scale of vertical mixing, because: 1) Down-welling of CH4 depleted air from the stratosphere. 2) The north-south gradient is increased when the CH4 lifetime reduces, influencing the exchange of methane in the tropical upper troposphere. A 10 ppb gradient is detectable, although transport model errors may result in similar or even larger differences, which is why we added the subsequent sentences indicating that in practice it is not easy to make use of this information.

*Technical corrections: Page 16, line 14: "To make" -> "making"*

Done.

*Page 16, line 16: "from" -> "for"*

Done.

*Figure 3: The use of a blue-red colorbar in this figure implies positive and negative contributions. You should consider changing to non-diverging colors.*

Done.
* * *
[Figure]

**Fig. 1.** Zonal mean differences in CH4 mixing ratio between a standard CH4 simulation and a simulation in which sources and sinks are both increased by 10% (difference= 10% Increase - Standard).

---

## Author Comment (AC3) · 11 Oct 2016

**Response to Anonymous Referee 3**

We would to thank the referee for the time and effort spent to help improve our manuscript. The structure of this document is as follows: Referee text's are in Italic font, answers are in Roman, modifications to the text are in bold font.

*General comments: The paper presents a comprehensive overview of inverse modeling of methane sources and sinks, describing the state-of-the-art of research in this field, including the 'historical' development, and giving future perspectives. The manuscript will also serve as an instructive introduction to the field. The paper is very well written and is recommended for publication after only few minor revisions.*

[Figure]

*Specific comments: Page 4, line 27: CH4 is not absent in the stratosphere, although concentrations are low and a steep vertical gradient exists. Are you referring to CH4 sources, which are absent in the stratosphere?*

The sentence was changed into: "**Low CH4 mixing ratios in the stratosphere matter,** ..."

*Although the authors refer to Rayner et al. (2016) for the notation of variables and terms of the equations, it would be helpful for the reader (and better suit the didactical purpose of the paper) to include the definitions directly in the paper.*

We followed the advise of the referee and added this table in a new Appendix B.

*Page 8, line 1: Isn't this the 13C analogue of the more general equation 2.*

Corrected (well spotted).

*Page 17, eq. A1, A3: For consistency subscripts should be used for M.*

Done.

*Technical comments: Page 5, line 31: ...(as in equation 1). ...the state vector components...*

Done.

*Page 16, line1: ... representation of wetland hydrology,...*

Done.

*Page 16, line 16: from -> for*

Done.

*Page 18, line 14: Röckmann*

Done.

*Page 19, line 18: Please check editor name.*

Done and corrected.

*Page 22, line 30: 2. Inverse modeling of CH4 fluxes...*

Corrected.

*Page 23, line 24-27: Please check spelling of co-author names*

Done and corrected.

*Figure 1: It could be helpful to indicate in the graphs whether isotopic information was used.*

This was in the figure already (green colors).

*Please specify for references 19 and 20 the scenario that was used in this graph.*

Done.

*Figure 3: ... January (left) and July (right). The middle and bottom panel show ... top panel.*

Corrected.

*Figure 3: Better not use a divergent color scale because in particular a blue-white-red color scale suggests negative and positive values or deviations from a central value.*

Done.

---

## Editor Comment (EC1) · M. Scholze (Editor) · 12 Oct 2016

Dear Sander,

Many thanks for providing your Author comments to the three referee comments.

It seems that in your comments to the third point raised by referee 2 the sentences clarifying the use of vertical profile information are missing. Could you please add them to your response and submit the revised manuscript accordingly.

Best regards

Marko
* * *